# Machine Learning Assisted Prediction of Airfoil Lift-to-Drag Characteristics for Mars Helicopter

Pengyue Zhao [1,2,3,*], Xifeng Gao [1,2,*], Bo Zhao [1,2], Huan Liu [1,2], Jianwei Wu [1,2] and Zongquan Deng [3]

1   Center of Ultra-Precision Optoelectronic Instrumentation Engineering, Harbin Institute of Technology, Harbin 150001, China
2   Key Laboratory of Ultra-Precision Intelligent Instrumentation, Ministry of Industry Information Technology, Harbin 150080, China
3   State Key Laboratory of Robotics and System, Harbin Institute of Technology, No. 92, Xidazhi Street, Nangang District, Harbin 150001, China
*   Correspondence: pyzhao@hit.edu.cn (P.Z.); 20220246@hit.edu.cn (X.G.); Tel.: +86-0451-86412041 (P.Z.)

**Abstract:** The aerodynamic properties of rotor systems operating within low Reynolds number flow field conditions are profoundly influenced by their geometric and flight parameters. Precise estimation of optimal airfoil parameters at different angles of attack is indispensable for enhancing these aerodynamic properties. This study presents a technique for optimizing the airfoil parameters of a Mars helicopter by employing machine learning methods in conjunction with computational fluid dynamics (CFD) simulations, thereby circumventing the need for expensive experiments and simulations. The effectiveness of diverse machine learning algorithms for prediction is evaluated, and the resultant models are utilized for airfoil optimization. Ultimately, the aerodynamic properties of the optimized airfoil are experimentally validated. The experimental findings exhibit agreement with the simulated predictions, indicating the successful optimization of the aerodynamic properties. This research offers valuable insights into the influence of airfoil parameters on the aerodynamic properties of the Mars helicopter, along with guidance for airfoil optimization.

**Keywords:** machine learning; Mars helicopter; computational fluid dynamics; airfoil; aerodynamic properties

## 1. Introduction

The exploration of Mars is of significant importance for understanding the origins and evolution of the planets in the solar system, discovering extraterrestrial energy sources, and expanding the potential living environments for humanity [1,2]. Due to the thin atmosphere near the surface of Mars, Mars helicopters have gained widespread attention from research institutions around the world for their ability to perform vertical take-off and landing, as well as high-altitude flight observations of the planet's rugged terrain [3], as shown in Figure 1. However, with the atmospheric density on Mars being only 1/70 of Earth's and the surface air pressure being only 1/100, Mars helicopters operating in low Reynolds number flow fields will experience reduced flight efficiency due to viscous effects and laminar separation on the helicopter's blades [4]. Therefore, it is crucial to study the aerodynamics of Mars helicopter rotor systems, such as airfoil shape and blade properties, under Martian atmospheric conditions.

Research on Mars helicopter rotor systems includes both numerical simulations and ground experiments. In numerical simulations of airfoil lift-drag characteristics, it has been found that the geometric and flight parameters of the airfoil can significantly affect the aerodynamic properties of the rotor system under low Reynolds number flow conditions. Unconventional airfoil structures with larger camber and lower maximum thickness can considerably improve the lift coefficient and lift-to-drag ratio. Studies by Takaki et al. [5] have shown that the Reynolds number in the flow field is strongly correlated with the

lift-to-drag ratio of the airfoil, whereas the Mach number is strongly correlated with the airfoil's drag coefficient. Furthermore, geometric parameters such as the maximum camber position and maximum camber can significantly influence the lift-drag characteristics of airfoils under low Reynolds number flow conditions ($Re < 10^5$) [6]. Srinath et al. [7] considered the combined effects of airfoil geometry and flight parameters on lift-drag characteristics and identified low Reynolds number airfoils with higher lift coefficients and lift-to-drag ratios. Selig et al. [8] compared various low Reynolds number airfoils under low Reynolds number and high Mach number flow conditions, finding that the E387 airfoil is more suitable for the Martian atmospheric flow field. Lin et al. [9] discovered that the high camber structure of the E387 airfoil can effectively suppress laminar separation on its surface and improve the lift-to-drag ratio. Spedding et al. [10] used numerical simulations to optimize the leading and trailing edges of low Reynolds number airfoils, effectively improving their lift-drag characteristics.

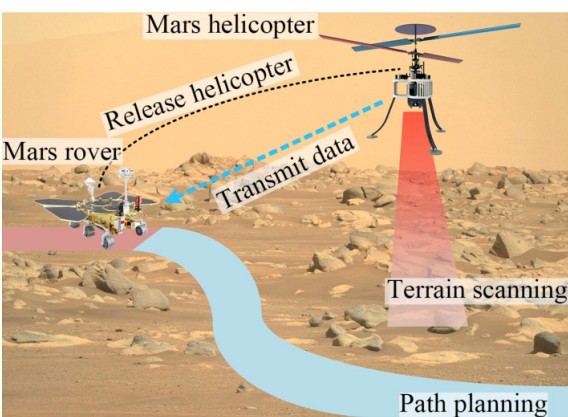

**Figure 1.** Schematic diagram of terrain scanning and path planning mission using Mars helicopter.

In the study of rotor blade geometry under low Reynolds number conditions, normally it is necessary to increase the blade pitch angle to generate sufficient hovering thrust. However, a larger pitch angle can cause vortices to form at the leading and trailing edges of the blade, resulting in energy loss and reduced aerodynamic efficiency [11–13]. Under the same pitch angle, increasing the blade's aspect ratio can improve the lift coefficient during hover and reduce the drag coefficient, thereby achieving higher aerodynamic efficiency [14]. When the blade's aspect ratio increases to 3.0, the effect of the Reynolds number on lift and drag coefficients becomes less significant, making it difficult for the aspect ratio to significantly influence the rotor system's aerodynamic performance [15]. Jan et al. [16] found that under low pitch angle conditions, a high aspect ratio can improve the flight efficiency of rotor systems and reduce the power consumption of rotor systems. Increasing the blade pitch angle can cause additional vortices to form at the leading edge of rotor blades with low aspect ratios, thus increasing thrust and suppressing rotor stall. Hassanalian et al. [17] employed strip theory to establish an aerodynamic model for rotor blade geometry and utilized numerical simulation methods to optimize blade geometric parameters and flight parameters. They employed polynomial functions to simulate the flow characteristics at the leading and trailing edges, and through theoretical methods, verified the optimization model for the wing span. They discovered that altering the blade geometry could significantly enhance the blade's thrust and aerodynamic efficiency. Leishman et al. [18] proposed an aerodynamic model of the coaxial rotor system for flight conditions on Mars and verified it through experiments, finding that the low Reynolds number flow condition on Mars would lead to a significant reduction in aerodynamic efficiency for rotor systems with conventional airfoils. Bohorquez et al. [19] conducted simulations and experiments on rotor systems with Reynolds numbers ranging from $5 \times 10^3$ to $6 \times 10^4$, revealing that airfoil shapes with small round leading edges and sharp trailing edges are more suitable for low Reynolds number flow fields. Benedict et al. [20]

employed PIV methods to measure the surface flow field of rotors in hovering states with Reynolds numbers not exceeding $3 \times 10^4$, discovering that airfoils with thin thicknesses and a camber range of 4.5% to 6.5% have higher aerodynamic efficiency. Additionally, increasing the twist angle of the rotor blade (from –10° to –20°) could effectively improve aerodynamic efficiency. Shrestha et al. [21] conducted hovering experiments for rotor systems under simulated Martian atmospheric conditions, obtaining the action rules of the Reynolds number and Mach number on the rotor system's aerodynamic efficiency *FM*, finding that selecting the optimal rotor airfoil could effectively increase the rotor system's aerodynamic efficiency *FM* from 0.34 to 0.6.

Due to the high costs of experiments and the immense computational demands of simulations, machine learning methods are increasingly used in airfoil optimization to enhance aerodynamic performance. Song et al. [22] introduced a machine learning optimization algorithm to improve airfoil performance, validating the predicted aerodynamic performance with experimental data. The findings demonstrate that machine learning is more computationally efficient than conventional genetic algorithms for optimizing lift-to-drag characteristics. Du et al. [23] established a deep-learning-based convolutional neural network framework (DPCNN) for airfoil design and performance optimization. They optimized aerodynamic performance parameters using the gradient descent method, achieving airfoil database parameterization and performance prediction with superior robustness and convergence. Li et al. [24] developed a nonstationary aerodynamic reduced-order model employing a long short-term memory (LSTM) network within a machine-learning approach. They parameterized the maximum camber and maximum thickness of the NACA6 series airfoil, applying it to the uncertainty and sensitivity analysis of geometric parameter variations. Zhang et al. [25] created a convolutional neural network (CNN) model for variable flow field, investigating lift coefficients for different airfoil geometry, Mach numbers, Reynolds numbers, and angles of attack. Additionally, they compared the prediction accuracy of a multilayer perceptron (MLP) with that of a CNN, demonstrating that the CNN exhibits superior prediction accuracy with minimal constraints in geometric representation. Zhu et al. [26] examined high Reynolds number turbulence around airfoils. Computational fluid dynamics (CFD) simulation results were employed as a dataset to construct a machine-learning-based model, incorporating different models for distinct regions of the flow field. The results indicated that the model could predict the flow state of the airfoils, with the prediction outcomes consistent with the original Spalart–Allmaras (SA) model calculations. Besides, the multiobjective optimization method by Wang et al. [27,28] has potential value to the application of Mars helicopter.

In this work, the flow fields of different NACA airfoils in the Mars atmosphere are simulated based on computational fluid dynamics, and the effects of airfoil camber, maximum camber position, thickness, and angle of attack on the lift-to-drag characteristics of the airfoils are investigated separately. Moreover, by exploring different machine learning algorithms for prediction, evaluating the prediction effect of different algorithms and then applying the model to optimize the airfoil, and using the optimized airfoil for experiments in the simulated Martian environment, we provide guidance to elucidate the effect of airfoil parameters on the lift-to-drag characteristics and airfoil optimization.

## 2. Simulation and Machine Learning Methods

### 2.1. Two-Dimensional CFD Simulation Methods

In this study, we primarily focus on NACA airfoils, with the schematic diagram of the airfoil geometry and its geometric parameters shown in Figure 2. The upper and lower curve equations for NACA airfoils are presented in Equation (1). NACA airfoils include parameters such as the airfoil's relative camber (*m*), the position of the maximum camber (*p*), and the relative thickness (*t*). The airfoil's relative camber is the ratio of the maximum camber to the chord length, whereas the position of the maximum camber is the ratio of the distance from the airfoil's leading edge to the maximum camber location to the chord

length. The airfoil's relative thickness is the ratio of the airfoil's maximum thickness to the chord length.

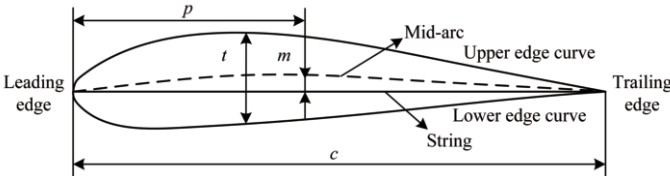

**Figure 2.** Schematic diagram of airfoil geometry and geometric parameters.

$$y_{\mathrm{c}} = \begin{cases} m\dfrac{x}{p^2}(2p - \dfrac{x}{c}), 0 \leq x \leq pc \text{ (upper edge curve)} \\ m\dfrac{c-x}{(1-p)^2}(1 + \dfrac{x}{c} - 2p), pc \leq x \leq c \text{ (lower edge curve)} \end{cases} \quad (1)$$

In this study, we use the finite element analysis software Fluent to carry out numerical simulations of the flow field around NACA airfoils under Martian atmospheric conditions. We investigate the effects of geometric parameters such as relative camber, maximum camber position, and relative thickness, as well as flight parameters such as rotor speed and angle of attack on the lift-to-drag characteristics of the airfoils. The aim is to comprehensively compare low-Reynolds-number NACA airfoils with excellent aerodynamic performance. Considering the influence of airfoil camber, maximum camber position, and relative thickness on the aerodynamic characteristics, special refinement of the grid in the leading and trailing edge regions is required. The airfoil grid range is 40 times its effective size, and the two-dimensional grid of the airfoil for finite element analysis is shown in Figure 3.

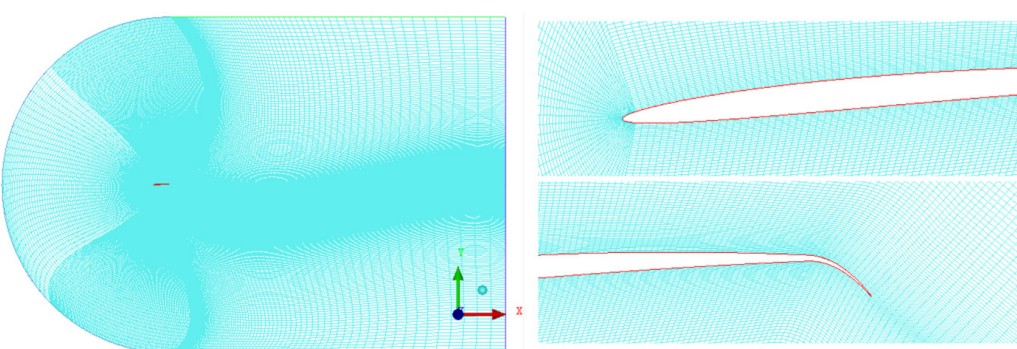

**Figure 3.** Mesh used in the 2D CFD calculation of NACA airfoils.

Additionally, taking into account the compressibility of the Martian atmospheric flow field, the airfoil finite element simulation uses a density-based solver and the SST $k$–$\omega$ model for calculations. The Reynolds number of the flow field around the Martian ranges from $10^4$ to $10^5$, which causes the laminar flow in the airfoil boundary layer to be easily disturbed and transformed into turbulent flow. The laminar flow separation and laminar-turbulent transition mechanism of airfoil flow in the Martian atmosphere has been discussed in detail in work [29,30]. Thus, the SST $k$–$\omega$ model is employed to simulate the turbulent flow on the airfoil surface in this paper, which is also used in the literature [31] to conduct the Martian airfoil aerodynamic performance simulation. The airfoil dimensions are consistent with actual conditions, and the flight parameters, such as flow velocity and angle of attack, are consistent with the experiments. The simulation parameters for the Martian atmosphere are shown in Table 1.

**Table 1.** Numerical simulation parameters for Mars atmospheric environment.

| Parameters | Values |
| --- | --- |
| Gas density (kg/m$^3$) | 0.0167 |
| Constant pressure specific heat capacity (J/kg·K) | 831.2 |
| Thermal conductivity (W/m·K) | 0.0132 |
| Gas viscosity (kg/m·s) | $1.289 \times 10^{-5}$ |
| Reference viscosity (kg/m·s) | $1.289 \times 10^{-5}$ |
| Reference temperature (K) | 210 |
| Effective temperature (K) | 260 |
| Molar mass (g/mol) | 44 |
| Reference area (m$^2$) | 0.04 |
| Reference length (m) | 0.04 |
| Reference depth (m) | 1 |

The airfoil surface boundary conditions are set as wall boundary conditions. Considering that the fluid environment is compressible and the flow field boundary distance is much larger than the distance between the airfoils, the inlet boundary condition is set as a pressure far-field boundary condition. The flow field at the exit of the airfoil trailing edge is set as a pressure outlet boundary condition. The pressure and temperature parameters for each boundary condition are consistent with the physical parameters of the Martian atmosphere. Specifically, the airfoil surface serves as an internal boundary with a wall boundary condition, whereas the circular arc boundary in front of the airfoil, as well as the upper and lower linear boundaries, are external boundaries with pressure far-field boundary conditions. The rear linear boundary is an external boundary with a pressure outlet boundary condition.

### 2.2. Three-Dimension CFD Simulation Methods

In this study, a three-dimensional simulation of the Mars helicopter rotor system is also carried out to investigate the aerodynamic characteristics. Considering the complex geometry of the rotor system, an unstructured mesh is used to divide the finite element flow field near the rotor. This unstructured mesh is in the form of a cylinder, with a diameter of 1.1 times the rotor blade diameter (*D*) and a height of 0.4*D*. Additionally, this mesh is set as a rotating sliding mesh in the finite element calculations, with its rotation center coinciding with the rotor axis. The unstructured mesh is shown in Figure 4a. A structured mesh is used to divide the remaining parts of the rotor system. This structured mesh is also in the form of a cylinder, with both its diameter and height being 3.0*D*. The structured mesh is shown in Figure 4b.

(a)　　　　　　　　　　　　　　　　　　　(b)

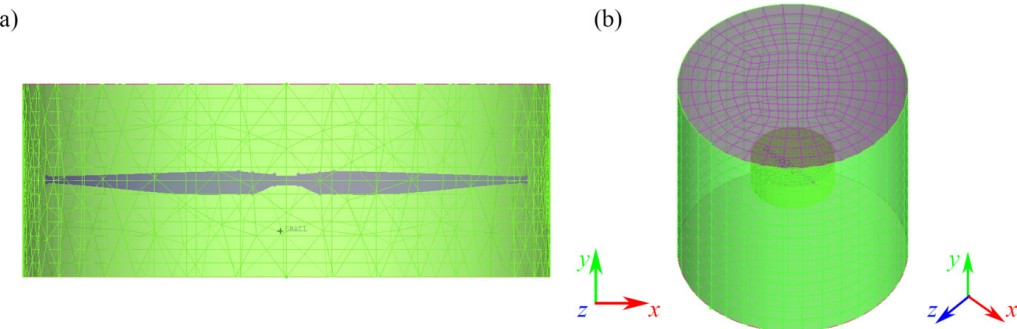

**Figure 4.** Mesh used in the 3D CFD calculation of the rotor system. (**a**) Unstructured grid around rotor blades (**b**) Structured grid of the rotor periphery.

The boundary conditions for the rotor blade surface, the upper boundary of the outer flow field, the lower boundary of the outer flow field, and the surrounding boundary of the outer flow field are all set as wall boundary conditions. The upper boundary, lower boundary, and surrounding boundary conditions where the flow field near the rotor intersects the outer flow field are set as coupled boundary conditions. The pressure parameters for all boundary conditions are set to 640 Pa, and the flow field temperature is set to 210 K, with simulation parameters consistent with the physical parameters of the Martian atmosphere. The rotor blade surface (internal boundary) and the upper, lower, and surrounding boundaries of the outer flow field (external boundary) are set as wall boundary conditions, whereas the upper, lower, and surrounding boundaries of the inner flow field (internal boundary) are set as coupled boundary conditions.

### 2.3. Machine Learning Models

Following the airfoil simulation, a dataset is generated based on the simulation results. The input parameters encompass airfoil parameters $m$, $p$, $t$, $\alpha$, whereas the output parameters include $C_l/C_d$ and $C_l^{1.5}/C_d$. The influence of airfoil parameters on lift-to-drag characteristics is examined through machine learning training. Simulation results serve as the basis for model training in machine learning, with different algorithmic models being trained on scikit-learn platforms. These regression algorithms comprise AdaBoost, support vector machine (SVM), and artificial neural network (ANN). ANN, also known as a multilayer perceptron (MLP), is a supervised learning algorithm suitable for the regression of nonlinear functions with multiple feature values [32]. A typical ANN structure consists of input, hidden, and output layers. In contrast to logistic regression, ANN incorporates one or more nonlinear layers, i.e., hidden layers [33]. ANN possesses the ability to learn nonlinear models in real time. However, a drawback is that different random weight initializations can result in additional validation accuracy. Support vector machine (SVM) is a supervised learning method that excels in high-dimensional spaces and scenarios where the number of dimensions exceeds the sample count [34]. The SVM regression in this study consists of Gaussian-SVM and Linear-SVM, depending on the Gaussian and linear kernel functions. SVM regression with a Gaussian kernel function, also referred to as radial basis function (RBF), necessitates considering hyperparameters gamma and $C$ [35,36]. Parameter $C$ influences the decision surface smoothness and the regression accuracy in the data samples. Meanwhile, the gamma parameter determines the impact of individual training examples. The grid search method is frequently employed for proper hyperparameter settings. The linear kernel function is defined as Equation (2):

$$K(x,y) = x \cdot y \tag{2}$$

where the $x$ and $y$ mean the sample $x$ and $y$. $K(x,y)$ is the value obtained by returning a new sample after calculation. The Gaussian kernel function is defined as Equation (3):

$$K(x,y) = e^{-\gamma\|x-y\|^2} \tag{3}$$

where the $\gamma$ means the hyperparameter in the kernel function.

The AdaBoost algorithm is a meta-estimator that initially fits the dataset, constructs an original regressor, and then adjusts the weights in subsequent regressors based on the current prediction error [37]. The weights are continuously refined until a predefined sufficiently small error rate is achieved or a predetermined maximum number of iterations is reached to determine a strong classifier [38]. In this study, the strong regressor within the AdaBoost model can be represented as [39], as shown in Equation (4):

$$y(x) = \sum_{m=1}^{M} \alpha_m G_m(x) \tag{4}$$

where $G_m(x)$ represents the weak regressor obtained in the $m$ iteration, and $\alpha_m$ represents the weight of $G_m(x)$ in the final regressor.

In this work, airfoil optimization comprises three primary stages. First, an airfoil database is established. The airfoil function is obtained by inputting airfoil parameters through *C* programming, and the lift-to-drag characteristics with the parameters to be optimized ($C_l/C_d$ and $C_l^{1.5}/C_d$) are acquired via 2D-CFD simulation. Next, the dataset is trained using various machine learning algorithm approaches, and the best regression algorithm is determined based on the prediction results ($MSE$ and $R^2$) to optimize airfoil parameters. Lastly, the rotor is redesigned according to the optimization results, and the outcomes of the airfoil optimization are validated through 3D-CFD simulations and experiments, as shown in Figure 5.

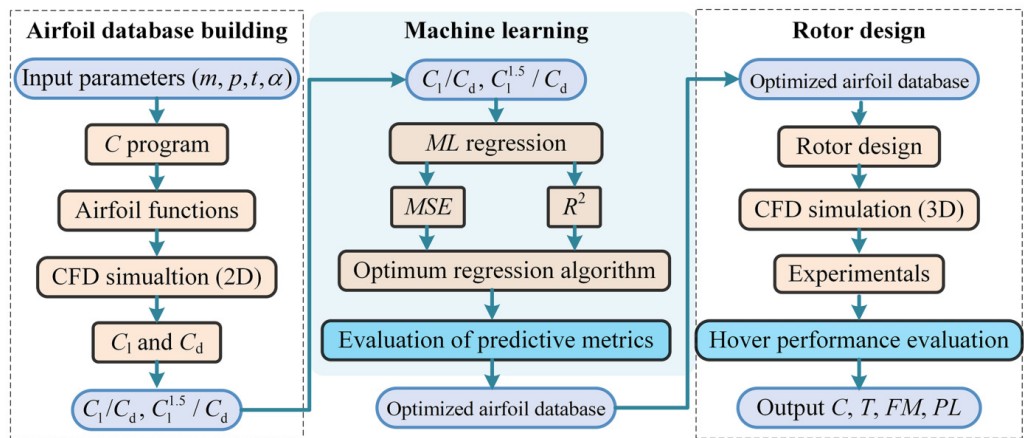

**Figure 5.** Airfoil optimization and verification main process.

## 3. Results and Discussion

### 3.1. FEM Simulations

$C_l/C_d$ and $C_l^{1.5}/C_d$ are crucial parameters for investigating the lift-to-drag characteristics of Mars helicopters. The primary objective of optimizing these characteristics is to identify the airfoil geometric parameters that correspond to their maximum values. Lift-to-drag characteristics of various NACA airfoils are computed using CFD simulations, with parameter $m$ ranging from 4 to 7, parameter $p$ from 6 to 8, and parameter $t$ from 4 to 6, as illustrated in Figure 6. In numerical simulations and CFD simulations, the finite element simulation is calculated until the residual is less than $10^{-5}$, at which point the lift resistance coefficient converges to a stable value, then the calculation is considered converged. The simulation results reveal that the $C_l/C_d$ decreases as parameters $m$ and $t$ increase, whereas it improves with an increase in parameter $p$. The trend exhibited by the $C_l^{1.5}/C_d$ curves in the graph is not immediately apparent, necessitating further statistical analysis to draw conclusions. Since the Mars atmosphere is complex, and the input parameters of the 3D simulation are more than those of 2D simulation, 3D CFD simulation will greatly increase the computation time and convergence difficulty, and it still has an influence on machine learning calculation. Note that the 3D structure of the rotor can also be obtained by 3D simulation, but considering the time cost of mesh drawing and simulation calculations, this work focuses on using a combination of 2D airfoil simulation and machine learning to optimize the design of the airfoil structure.

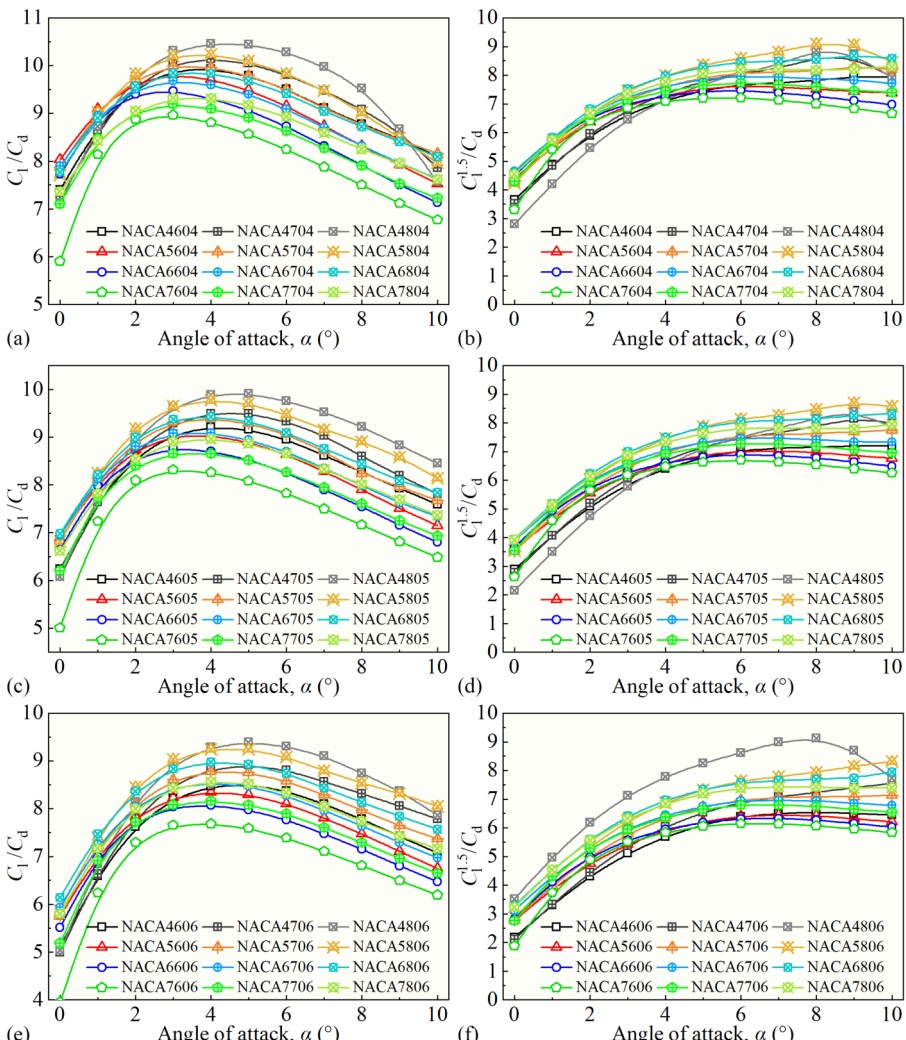

**Figure 6.** Relationship between lift-to-drag characteristics and angle of attack of different airfoils: (**a**,**b**) the relative thickness is 4%. (**c**,**d**) The relative thickness is 4%. (**e**,**f**) The relative thickness is 6%.

### 3.2. Regression Results of Different Algorithms

To further examine the influence of different parameters on lift-to-drag characteristics, machine learning methods are employed to carry out regression and prediction based on existing datasets. The samples are from the original dataset, and the training/testing samples are both from the same set. The dataset is split into training and testing sets which the training set accounts for 80% of all samples and the test set accounts for 20%. The input dataset is separately predicted in different models trained using AdaBoost, SVM, and ANN algorithms based on the CFD simulations. When the number of layers of multilayer perceptron increases, the regression result of ANN 40-40-40 of $C_l^{1.5}/C_d$ has an *MSE* of 0.034 and $R^2$ of 0.966. The regression result decreases compared to the original structure. In order to avoid overfitting and the degradation of the nonlinear fitting effect, the number of network layers is not increased in this work. The dataset has been standardized scaling all data to between $(-1, 1)$ before split into the training and testing sets. The predicted values are compared with the original output results in the dataset. The visualized results of $C_l/C_d$ are illustrated in Figure 7, which represents the degree of deviation between the predicted values and actual results across various models. For the $C_l/C_d$ parameters, in the prediction results of SVM-L and several ANN regression algorithms, most of the points deviate from the diagonal line in the plot, with only a few points clustering near the diagonal line. This indicates a significant deviation in the prediction results. Furthermore, it can be observed that increasing the number of nodes in the hidden layer of the ANN

algorithm does not significantly affect the overall prediction trend. In contrast, during the SVM-L algorithm procedure, the optimal hyperparameters within the range have been determined before performing the final data prediction. Therefore, SVM-L and ANN are not suitable for predicting $C_l/C_d$ parameters and searching for the best fit. The prediction results of AdaBoost and SVM-G require further evaluation.

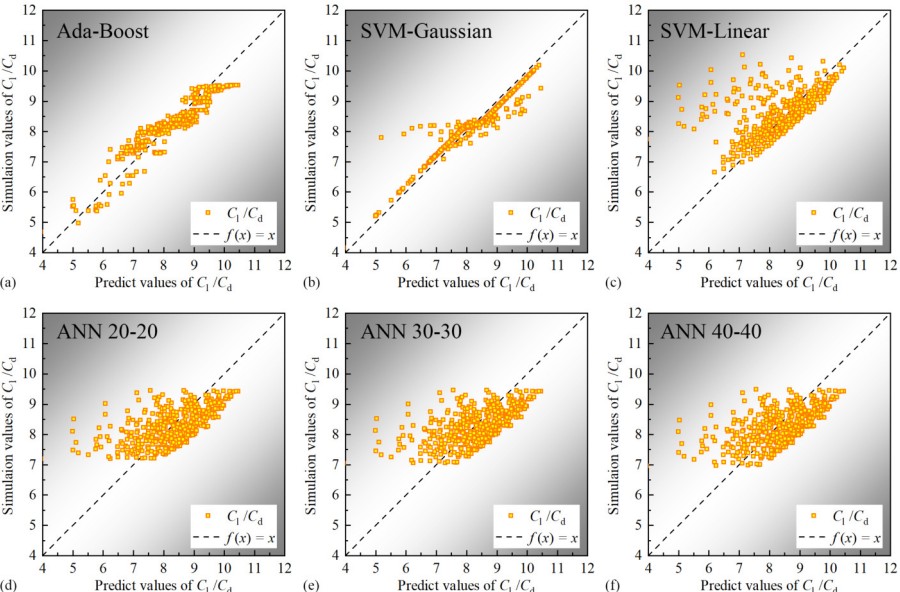

**Figure 7.** $C_l/C_d$ regression results of different algorithms.

The visualized results of $C_l^{1.5}/C_d$ prediction are depicted in Figure 8, demonstrating that different models exhibit varying applicability for distinct parameters. The predictions of the SVM-L algorithm significantly deviate from the original output values in the dataset, with a large number of points straying from these values. For the SVM-G algorithm, most predicted values closely align with the actual simulated values, except for a few points concentrated around 3–5 and 9. The predictions of ANN and AdaBoost regression algorithms for $C_l^{1.5}/C_d$ do not significantly deviate from the actual simulated values, necessitating further error evaluation.

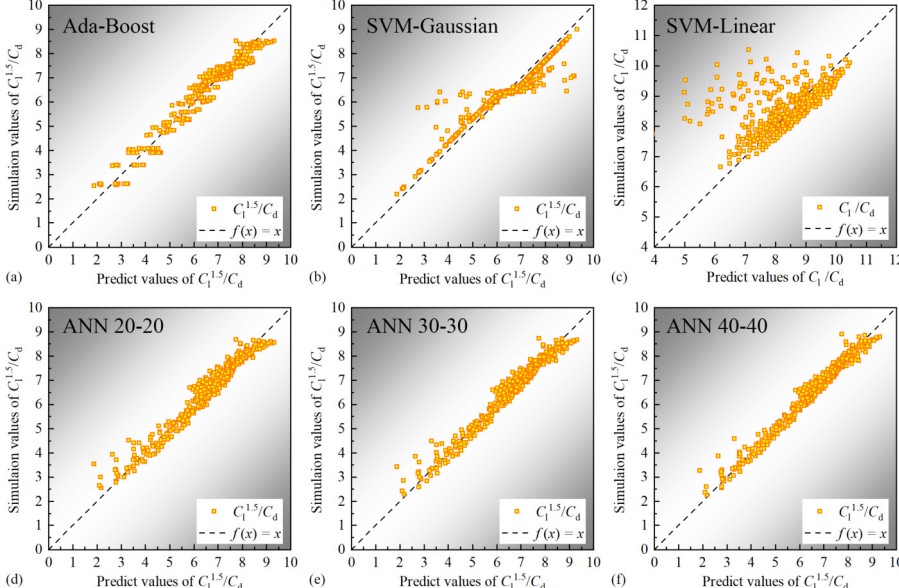

**Figure 8.** $C_l^{1.5}/C_d$ regression results of different algorithms.

*3.3. Evaluation of Different Algorithms*

In machine learning, regression algorithms are assessed based on mean square error (*MSE*) [40] and coefficient of determination ($R^2$) [41] when making predictions. The mean square error represents the expected value of the squared error, with the error being the difference between the estimated and actual values. The MSE expression is shown in Equation (5):

$$MSE = \frac{1}{n} \sum_{i=1}^{n} (y_i - \hat{y}_i)^2 \tag{5}$$

where $y_i$ denotes the actual value, $\hat{y}_i$ is the predicted value, and $n$ represents the total number of samples. A smaller *MSE* indicates that the predicted value of the dataset is closer to the actual value, although *MSE* faces the issue of scale variation. The $R^2$ score reflects the correlation between predicted and actual values of the model, distinguishing it from others and eliminating the influence of scale. The $R^2$ expression is provided in Equation (6):

$$R^2 = 1 - \frac{\sum_{i=1}^{n} (y_i - \hat{y}_i)^2}{\sum_{i=1}^{n} (y_i - \bar{y})^2} \tag{6}$$

where $\bar{y}$ is the average of the predicted values, and $n$ represents the total number of samples. $R^2$ scores range from 0 to 1, with the model's accuracy being higher when the value is closer to 1.

The integrated statistics for the MSE and R2 scores of $C_l/C_d$ and $C_l^{1.5}/C_d$ are presented in Figure 9. Regarding the MSE evaluation, the $C_l/C_d$ MSE reveals that the SVM-G algorithm offers superior prediction performance. The MSE for the ANN algorithm does not exhibit significant changes as the number of nodes in the hidden layer increases, and the SVM-L algorithm's *MSE* is considerably larger compared to other regression algorithms, as demonstrated in Figure 9a. This observation aligns with the visualization results in Figure 6. For the $C_l^{1.5}/C_d$ prediction *MSE*, the ANN algorithm's *MSE* is notably lower than that of other regression algorithms. It gradually decreases as the number of nodes in the hidden layer increases, indicating improved prediction accuracy, as illustrated in Figure 9b. In line with the *MSE* evaluation results, the $C_l/C_d$ parameters exhibit the highest prediction accuracy with the SVM-G algorithm, followed by AdaBoost, whereas SVM-L has the highest $R^2$ value. Moreover, the $R^2$ value of the ANN algorithm progressively increases with the number of nodes in the hidden layer, as depicted in Figure 9c. The $R^2$ values of $C_l^{1.5}/C_d$ parameters reflect the varying prediction results of different algorithms, ranging from excellent to poor. The ANN algorithm boasts a high $R^2$ value that increases with the number of nodes in the hidden layer, followed by AdaBoost and SVM-G. Conversely, the SVM-L algorithm has the lowest $R^2$ value, suggesting that its prediction results deviate significantly from the sample values.

*3.4. NACA Airfoil Optimization and ML Prediction*

After evaluating the prediction results from various machine learning regression algorithms, the best-performing algorithm was chosen for new predictions. The SVM-G regression algorithms are used for $C_l/C_d$ and the ANN 40-40 is used for $C_l^{1.5}/C_d$ according to the *MSE* and $R^2$ score. The maximum values of $C_l/C_d$ and $C_l^{1.5}/C_d$ parameters at different angles of attack, along with the corresponding NACA airfoils, were initially predicted, as shown in Tables 2 and 3. Subsequently, finite element simulations were performed for the corresponding airfoils at different angles of attack to verify the accuracy of the model predictions. In the $C_l/C_d$ parameter-optimized airfoil parameters, it can be observed that the parameter m values are higher at the maximum and minimum $\alpha$ and around 4 in the intermediate range. Conversely, the parameter $p$ exhibits the opposite trend, with smaller values near 0 and 10 degrees and larger $p$ values in the intermediate range. The parameter $t$ appears optimal at 4 for angles of attack from 0 to 8 degrees and increases when $\alpha$ is between 9 and 10 degrees. Regarding the $C_l^{1.5}/C_d$ parameter, the interrelationship between the *m* and *t* parameters and $\alpha$ is similar to that of the $C_l/C_d$ parameter. However,

the parameter $p$ trend is approximately increasing slowly as $\alpha$ increases. The simulation results indicate that the prediction error for the $C_l/C_d$ parameter regression algorithm is within 2%, and the error for the $C_l^{1.5}/C_d$ parameter regression algorithm is within 3%. Meanwhile, airfoil parameters under different $\alpha$ are optimized, and the optimized $C_l/C_d$ and $C_l^{1.5}/C_d$ parameters show improvement compared to the maximum values in the original samples. The rotor structure has a twist angle of $-10°$ and the change of angle is linear along the spanwise direction, as shown in Figure 2. Along the spanwise direction, the rotor is divided into 20 sections, and the twist angle of each cross-section corresponds to parameter $\alpha$. The shape for each section is optimized according to the parameters $m$, $p$, $t$ which determine the section shape, as shown in Figure 2. The geometrical parameters of the two-dimensional airfoil for each cross-section are derived from the optimization results in Table 4, which means that we select the optimized airfoil parameters $m$, $p$, $t$ which determine the section shape according to the angles $\alpha$ of different cross-sections. The SVM-G regression algorithms are used for $C_l/C_d$ and the ANN 40-40 is used for $C_l^{1.5}/C_d$ according to the $MSE$ and $R^2$ score. Finally, the three-dimensional structure of the Mars helicopter rotor with airfoil shape variation is obtained by the envelope method with the best lift-drag characteristics. The AOA of the rotor varies linearly at different sections and the final envelope of the airfoil is obtained by optimizing the section parameters at the corresponding AOA.

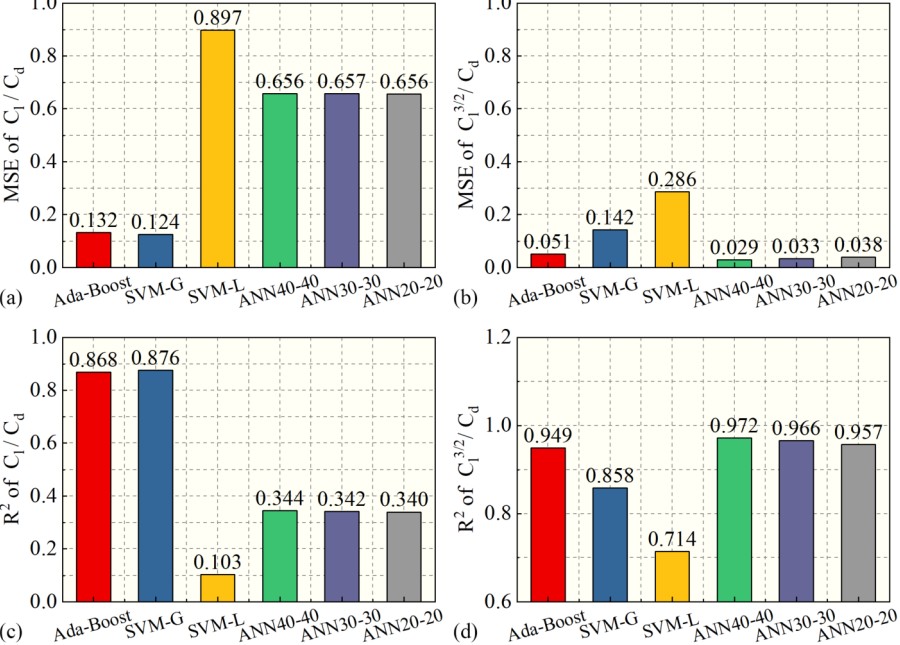

**Figure 9.** Evaluation of different regression algorithms: (**a**,**b**) $MSE$. (**c**,**d**) $R^2$.

After optimizing the airfoil shape at angles of attack from 0 to 10 degrees, the lift-to-drag characteristics at the interpolated angle of attack are predicted and validated by simulation using machine learning regression algorithms, as displayed in Tables 4 and 5. As the angle of attack gradually increases from 0.5 to 9.5 degrees, parameters $m$ and $p$ follow the same trend as in the original angle of attack interval. The parameter $m$ is smaller in the middle range of the $\alpha$ and larger at the marginal $\alpha$, whereas parameter $p$ exhibits the opposite trend. However, as the angle of attack gradually increases, the parameter $t$ of the optimal airfoil also increases from 4 to 4.25, similar to the trend of the $C_l/C_d$ parameter in the original interval. Simulations are conducted for the corresponding optimal airfoil types at different interpolation values of $\alpha$. The error between simulation and prediction results is calculated, revealing that the prediction error for the $C_l/C_d$ parameter regression algorithm is within 3%, and the error for the $C_l^{3/2}/C_d$ parameter regression algorithm is

within 5%. The prediction effect of the machine learning regression algorithm is closer to the actual simulation results.

**Table 2.** ML predicted $C_l/C_d$ maximum and corresponding airfoil type.

| $\alpha$ | $m$ | $p$ | $t$ | $Max_o$ | $Max_p$ | $Max_s$ | Relative Error (%) |
|---|---|---|---|---|---|---|---|
| 0 | 5.41 | 6.28 | 4.00 | 8.028 | 8.065 | 8.035 | 0.37 |
| 1 | 5.33 | 6.29 | 4.00 | 9.056 | 9.089 | 9.083 | 0.06 |
| 2 | 4.54 | 7.81 | 4.00 | 9.816 | 9.835 | 9.829 | 0.06 |
| 3 | 4.31 | 7.85 | 4.00 | 10.263 | 10.312 | 10.304 | 0.08 |
| 4 | 4.25 | 7.85 | 4.00 | 10.363 | 10.454 | 10.423 | 0.30 |
| 5 | 4.21 | 7.85 | 4.00 | 10.285 | 10.391 | 10.352 | 0.38 |
| 6 | 4.19 | 7.85 | 4.00 | 10.060 | 10.184 | 10.103 | 0.80 |
| 7 | 4.16 | 7.86 | 4.00 | 9.796 | 9.974 | 9.809 | 1.69 |
| 8 | 4.20 | 7.76 | 4.00 | 9.653 | 9.732 | 9.655 | 0.80 |
| 9 | 4.50 | 7.55 | 4.36 | 8.915 | 8.931 | 8.935 | −0.05 |
| 10 | 5.40 | 6.28 | 4.27 | 8.249 | 8.250 | 8.255 | −0.07 |

**Table 3.** ML predicted $C_l^{3/2}/C_d$ maximum and corresponding airfoil type.

| $\alpha$ | $m$ | $p$ | $t$ | $Max_o$ | $Max_p$ | $Max_s$ | Relative Error (%) |
|---|---|---|---|---|---|---|---|
| 0 | 6.48 | 7.65 | 4.00 | 4.644 | 4.650 | 4.652 | −0.05 |
| 1 | 6.51 | 7.61 | 4.00 | 5.822 | 5.829 | 5.836 | −0.12 |
| 2 | 6.29 | 7.76 | 4.00 | 6.773 | 6.816 | 6.777 | 0.57 |
| 3 | 5.98 | 7.84 | 4.00 | 7.465 | 7.525 | 7.481 | 0.58 |
| 4 | 5.34 | 7.89 | 4.00 | 7.934 | 7.988 | 7.961 | 0.33 |
| 5 | 4.64 | 7.93 | 4.00 | 8.311 | 8.361 | 8.316 | 0.54 |
| 6 | 4.39 | 7.95 | 4.00 | 8.576 | 8.617 | 8.612 | 0.06 |
| 7 | 4.32 | 7.94 | 4.00 | 8.797 | 9.000 | 8.813 | 2.12 |
| 8 | 4.31 | 7.87 | 4.00 | 9.315 | 9.396 | 9.338 | 0.63 |
| 9 | 5.25 | 7.89 | 4.00 | 9.160 | 9.161 | 9.189 | −0.31 |
| 10 | 6.22 | 7.91 | 4.00 | 8.881 | 8.886 | 8.904 | −0.20 |

**Table 4.** ML predicted $C_l/C_d$ maximum and corresponding airfoil type.

| $\alpha$ | $m$ | $p$ | $t$ | $Max_p$ | $Max_s$ | Relative Error (%) |
|---|---|---|---|---|---|---|
| 0.5 | 5.39 | 6.17 | 4.00 | 8.559 | 8.654 | −1.11 |
| 1.5 | 4.77 | 7.74 | 4.04 | 9.481 | 9.422 | 0.63 |
| 2.5 | 4.40 | 7.80 | 4.11 | 10.186 | 10.010 | 1.75 |
| 3.5 | 4.31 | 7.80 | 4.15 | 10.538 | 10.278 | 2.54 |
| 4.5 | 4.28 | 7.80 | 4.17 | 10.536 | 10.257 | 2.71 |
| 5.5 | 4.25 | 7.81 | 4.18 | 10.373 | 10.075 | 2.96 |
| 6.5 | 4.17 | 7.85 | 4.16 | 10.065 | 9.815 | 2.54 |
| 7.5 | 4.16 | 7.83 | 4.17 | 9.870 | 9.703 | 1.72 |
| 8.5 | 4.28 | 7.63 | 4.26 | 9.359 | 9.315 | 0.46 |
| 9.5 | 5.44 | 7.91 | 4.25 | 8.504 | 8.558 | −0.64 |

**Table 5.** ML predicted $C_l^{3/2}/C_d$ maximum and corresponding airfoil type.

| $\alpha$ | $m$ | $p$ | $t$ | $Max_p$ | $Max_s$ | Relative Error (%) |
|---|---|---|---|---|---|---|
| 0.5 | 5.71 | 6.98 | 4.00 | 5.188 | 5.216 | −0.54 |
| 1.5 | 5.96 | 8.00 | 4.00 | 6.341 | 6.343 | −0.03 |
| 2.5 | 5.94 | 7.86 | 4.00 | 7.047 | 7.168 | −1.68 |
| 3.5 | 5.96 | 7.85 | 4.09 | 7.683 | 7.722 | −0.50 |

**Table 5.** *Cont.*

| $\alpha$ | $m$ | $p$ | $t$ | $Max_p$ | $Max_s$ | **Relative Error** (%) |
|---|---|---|---|---|---|---|
| 4.5 | 4.62 | 7.87 | 4.12 | 8.126 | 8.063 | 0.79 |
| 5.5 | 4.47 | 7.86 | 4.15 | 8.557 | 8.327 | 2.77 |
| 6.5 | 4.31 | 7.88 | 4.15 | 8.828 | 8.517 | 3.65 |
| 7.5 | 4.29 | 7.85 | 4.17 | 9.347 | 8.933 | 4.63 |
| 8.5 | 4.44 | 7.77 | 4.25 | 9.473 | 9.201 | 2.96 |
| 9.5 | 5.74 | 7.81 | 4.25 | 9.232 | 8.995 | 2.63 |

## 4. Mars Atmospheric Environment Simulation and Experiment

### 4.1. Experimental Device

To simulate the Martian atmospheric environment, a Martian atmosphere simulator is constructed, with a cylindrical vacuum chamber having both a diameter and height of 3.0 m. Figure 10a shows the Martian atmosphere simulator (MAS), and Figure 10b shows the structure of the hovering test bench of the rotor system. Carbon dioxide gas is introduced into the vacuum chamber to simulate the Martian atmospheric gas composition, and the pressure within the chamber is adjusted by a Roots pump system to alter the Reynolds number range of the internal flow field. Using the Martian atmosphere simulator, the rotor system's experimental atmospheric density ($\rho$ = 0.0167 kg/m$^3$) is obtained to simulate the Martian atmospheric environment density. The gas composition inside the vacuum chamber is carbon dioxide, consistent with the Martian atmospheric environment. In hover experiments, the Reynolds number of the rotor system can be determined according to gas density and rotor speed. To ensure the stability of the test bench, a columnar structure is adopted for the hovering test bench of the rotor system, with a thrust measurement error of less than 0.01 N and a rotor power measurement error of less than 0.1 W.

Considering the wall effect of the vacuum chamber and based on its diameter of 3.0 m, the rotor diameter for experiments is determined to be 1.0 m. The rotor structure adopts a dual-blade configuration, as shown in Figure 11. The rotor has a twist angle of −10° along the spanwise direction. The airfoil data for each section is obtained from a machine-learning-optimized airfoil library, as shown in Table 4. The blades are made of carbon fiber material using 3D printing, which ensures sufficient strength and low mass for the rotor system due to the high centrifugal force acting on the blades during high-speed rotation. The blade edges are bonded with copper foil for trimming.

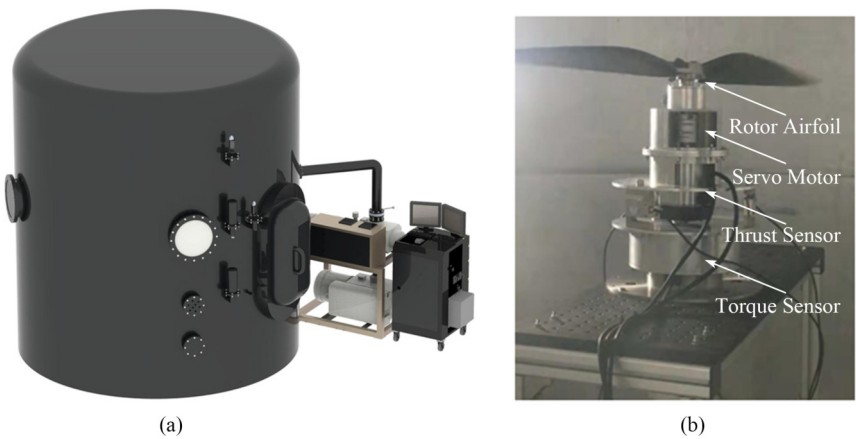

(a)　　　　　　　　　　　　　　　(b)

**Figure 10.** Experimental device for the rotor system of the Mars helicopter: (**a**) Martian atmosphere simulator. (**b**) Hovering test bench of the rotor system.

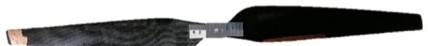

**Figure 11.** Structure of the low Reynolds number rotor.

Based on the trapezoidal blade geometry, the distribution curve of the blade chord length $c(\mathrm{r})$ along the spanwise direction can be obtained, as shown in Figure 12. According to the Martian atmospheric density and dynamic viscosity, the distribution curve of the Reynolds number, $Re$, for different positions of the rotor blade under a rotor speed $\Omega$ of 3000 r/min can be obtained.

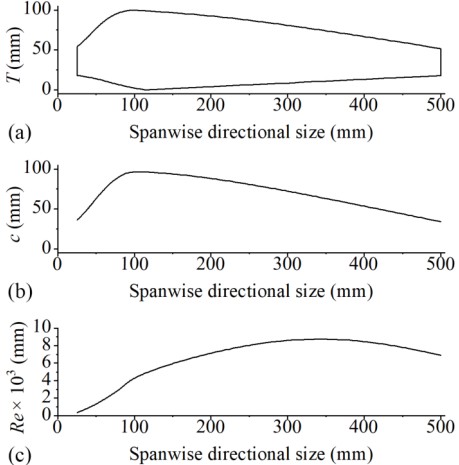

**Figure 12.** Geometrical parameters and characteristics of low Reynolds number single rotor blade: (**a**) Geometric shape. (**b**) Chord distribution. (**c**) Reynolds number distribution.

*4.2. Experiment Results*

Power loading ($PL$) and figure of merit ($FM$) are the main indices to evaluate the efficiency of a typical rotor system. $FM$ is a dimensionless quantity with the expression shown in Equation (7):

$$FM = \frac{P_{\text{idl}}}{P_{\text{actl}}} = \frac{P_{\text{idl}}}{P_{\text{i}} + P_0} = \frac{Tv_{\text{i}}}{P_{\text{tot}}} = \frac{C_{\text{T}}^{3/2}}{\sqrt{2} \cdot C_{P_{\text{meas}}}} < 1 \tag{7}$$

where $P_{\text{idl}}$ is the ideal power, and $P_{\text{actl}}$ is the actual power. $P_i$ is the induced power and the $P_0$ is the profile power which is related to the $C_l$ and $C_d$ of the airfoil respectively. $FM$ can be further expressed according to blade element momentum theory. $C_{\text{T}}$ is the thrust coefficient. In addition, $C_{P_{\text{meas}}}$ is defined as the experimental power coefficient. $PL$ characterizes the unit of thrust per unit power, the relationship between $PL$ and $FM$ is shown as Equation (8):

$$FM = \frac{Tv_{\text{i}}}{P_{\text{actual}}} = \frac{T}{P_{\text{actl}}} \sqrt{\left(\frac{T}{A}\right) \frac{1}{2\rho}} = PL \sqrt{\frac{DL}{2\rho}} \tag{8}$$

where the $v_{\text{i}}$ is the induced velocity; $A$ is the blade disk area; $\rho$ is the density of air; and $DL$ is the blade disk loading which is $T/A$.

The comparison of numerical simulation results and experimental results for the hover performance of a low Reynolds number rotor system under Martian atmospheric conditions is shown in Figure 13. Figure 13a–d respectively show the comparison of hovering thrust $T$, required power $P$, aerodynamic efficiency $FM$, and power loading $PL$ for the rotor system, where data points are taken at rotor speed intervals of 200 r/min within the range of 1000–3000 r/min. According to Figure 13, the trends of the predicted aerodynamic parameters for the rotor system are consistent with the experimental results. As the rotor

speed increases, the difference between the numerical simulation results and experimental results for rotor thrust and power becomes larger, as shown in Figure 13a,b. This also leads to the calculated aerodynamic efficiency and power loading results based on the numerical simulation being lower than the experimental results, as shown in Figure 13c,d. It is found that the error of numerical simulation for rotor thrust and power increases with the rotor speed, with the maximum error for thrust being approximately 14% and the maximum error for power being approximately 15%. This error is mainly due to the limitations of the SST $k$-$\omega$ model used in the numerical simulation for predicting the flow field distribution of laminar separation and separated bubbles on the blade surface in low Reynolds number flows. However, the maximum error of the numerical simulation results is within a reasonable range. Assuming the mass of the Mars helicopter to be 2.0 kg and that the thrust generated by the coaxial-rotor system is approximately 1.7 times that of the single-rotor system, it can be calculated that the minimum hovering thrust for the single-rotor system is 4.30 N, corresponding to a rotor speed of 2130 r/min and a power of approximately 83.8 W.

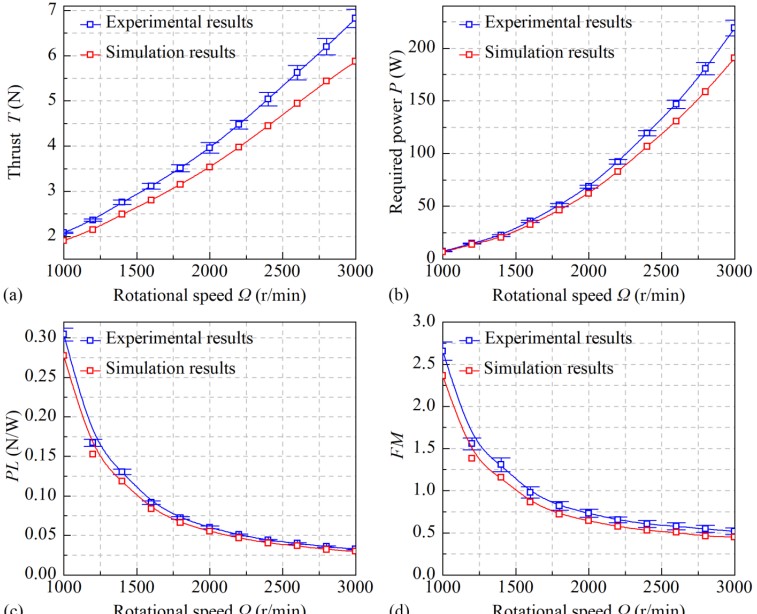

**Figure 13.** Comparison between numerical simulation results and experimental results of rotor hovering performance: (**a**) Generated thrust. (**b**) Required power. (**c**) Power Loading. (**d**) Figure of merit.

## 5. Conclusions

In this study, we propose a method for optimizing the airfoil parameters of a Mars helicopter using machine learning techniques applied to computational fluid dynamics simulations, thus circumventing costly experiments and simulations. The effectiveness of various machine learning algorithms for prediction is assessed, and the resulting models are utilized for airfoil optimization. The aerodynamic characteristics of the optimized airfoil are ultimately verified through experiments. The experimental results demonstrate consistency with the simulated predictions, indicating successful optimization of the aerodynamic characteristics. Key findings of this work include:

(1) Machine learning methods are suitable for predicting the aerodynamic characteristics of the rotor system of the Mars helicopter with varying geometric and flight parameters. The performance of different algorithms varies based on the output parameters, as indicated by mean squared error ($MSE$) and $R^2$ evaluations. The support vector machine linear (SVM-G) regression algorithm exhibits relatively high prediction accuracy. The SVM-G regression algorithms are used for $C_l/C_d$ and the ANN 40-40 is used for $C_l^{1.5}/C_d$ according to the $MSE$ and $R^2$ score.

(2) The discrepancies between the simulation and prediction results are assessed by calculating the errors. The prediction error for the lift-to-drag ratio ($C_l/C_d$) regression algorithm is within 3%, whereas the error for the $C_l^{3/2}/C_d$ parameter regression algorithm is within 5%. The prediction performance of the machine learning regression algorithm closely aligns with the simulation results.

(3) The comparison of hovering thrust ($T$), required power ($P$), aerodynamic efficiency ($FM$), and power loading ($PL$) results for the rotor system is conducted, with data points taken at rotor speed intervals of 200 r/min within the range of 1000–3000 r/min. The aerodynamic characteristics of the optimized airfoil are obtained through experiments. The experimental results are consistent with the predicted results of the simulation, validating the optimization of the aerodynamic characteristics.

**Author Contributions:** Conceptualization, P.Z.; Methodology, P.Z., Z.D.; Software, B.Z., J.W.; Validation, X.G.; Formal Analysis, B.Z., H.L.; Resources, B.Z., H.L.; Data curation, X.G.; Writing—original draft, P.Z., X.G.; Writing—review and editing, B.Z., H.L., J.W. and Z.D.; Supervision, P.Z.; Funding acquisition, P.Z., J.W. and Z.D. All authors have read and agreed to the published version of the manuscript.

**Funding:** This research was funded by the National Natural Science Foundation of China (52105547), the China Postdoctoral Science Foundation (2021M700995), and the Natural Science Foundation of Heilongjiang Province (LBH-Z21063).

**Institutional Review Board Statement:** Not applicable.

**Informed Consent Statement:** Not applicable.

**Acknowledgments:** We would like to express our gratitude to Zongquan Deng, Qiquan Quan, and Dewei Tang from Aerospace Mechanism Testing laboratory, HIT for their guidance and support in the field of Mars helicopter technology.

**Conflicts of Interest:** The authors declare no conflict of interest.

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
