# Peer review of "Machine Learning Assisted Prediction of Airfoil Lift-to-Drag Characteristics for Mars Helicopter"

_aerospace, doi:10.3390/aerospace10070614_

Round 1

Reviewer 1 Report

The manuscript presents a technique for optimizing the airfoil shape parameters of a Mars helicopter by employing machine learning methods, which reduces the cost of experiments and numerical simulations. The paper is overall well-written and the methods proposed are clearly presented. However, there are some major concerns which will be appreciated if the authors can elaborate more on.         

1. For section 3.2, in which the regression results of different methods are compared, are the samples from the original dataset? More specifically, did authors split the dataset into training/testing sets? Or the training and testing samples are both from the same set? The authors should clarify this to avoid ambiguity.                                                                   

2. Are there any data preprocessing done before training the model? For example, normalizing the data?                                                                

3. Have the authors looked at the training history of the ANN in the cl-cd model? it seems strange that the ANN performance saturates when we increasing the network size. I suggest try wider and deeper network to investigate the ANN effectiveness in this problem.                                               

4. In section 3.4, I suggest explicitly state that which regression algorithms  

are used for Cl-Cd and Cl^1.5/Cd.                              

5. As far as I'm concerned, the authors used the trained model to optimize the airfoil shape to achieve the best lift-drag characteristics. First of all, what kind of optimization algorithms was used? Moreover, I'm a little confused by the optimization setup. What is the purpose of this type of optimization? In my opinion, it fits best with morphing airfoils. For fixed shape ones, should we care more about the overall performance in the entire range, for example, AOA from 0 to 10 degrees, instead of the performance at a specific AOA?                                                 

6. Similar to the question just mentioned above, I assume that the authors did the experiments for the optimized airfoil shape. As the optimal shapes for different AOA are different, which shape is used in the experiments? Or the airfoil shape changes during the experiments.

7. In the conclusion, the authors stated that ``The support vector machine-linear (SVM-L) regression algorithm exhibits relatively high prediction accuracy.''  As far as I'm concerned, in neither the Cl-Cd nor the Cl^1.5/Cd model, SVM-L is not considered superior, I'm wondering how the author drew the conclusion. 

Author Response

Reviewer #1: The manuscript presents a technique for optimizing the airfoil shape parameters of a Mars helicopter by employing machine learning methods, which reduces the cost of experiments and numerical simulations. The paper is overall well-written and the methods proposed are clearly presented. However, there are some major concerns which will be appreciated if the authors can elaborate more on.

Point 1: For section 3.2, in which the regression results of different methods are compared, are the samples from the original dataset? More specifically, did authors split the dataset into training/testing sets? Or the training and testing samples are both from the same set? The authors should clarify this to avoid ambiguity.

Reply: Thank you for your suggestions. The samples are from the original dataset, and the training/testing samples are both from the same set. The dataset is split into training and testing sets which the training set accounts for 80% of all samples and the test set accounts for 20%. The description is clarified in the correspond section of the revised manuscript.

Change: The changes are marked as 5, which are highlighted in yellow color on Pages 7.

Point 2: Are there any data preprocessing done before training the model? For example, normalizing the data?

Reply: Thank you for your careful review. The dataset has been standardized scaling all data to between (-1,1) before split into the training and testing sets. The description is clarified in the correspond section of the revised manuscript.

Change: The changes are marked as 7, which are highlighted in yellow color on Page 7.

Point 3: Have the authors looked at the training history of the ANN in the cl-cd model? it seems strange that the ANN performance saturates when we increasing the network size. I suggest try wider and deeper network to investigate the ANN effectiveness in this problem.

Reply: Thank you for your suggestion. The current data set has a sample size of 400, and due to the large amount of CFD simulation calculations, increasing the number of neural network layers requires more data volume. When the number of layers of multilayer perceptron increases, the regression result of ANN 40-40-40 has an MSE of 0.034 and R2 of 0.966. The Cl1.5/Cd regression result decreases compared to the original structure. In order to avoid overfitting and the degradation of nonlinear fitting effect, the number of network layers is not increased in this work. In future work, we will further increase the depth of the network.

Change: The changes are marked as 6, which are highlighted in yellow color on Page 7.

Point 4: In section 3.4, I suggest explicitly state that which regression algorithms are used for Cl-Cd and Cl^1.5/Cd.

Reply: Thank you for your careful review. The SVM-G regression algorithms are used for Cl/Cd and the ANN 40-40 is used for Cl1.5/Cd according to the MSE and R2 score. The corresponding part are stated explicitly in section 3.4 of the revised manuscript.

Change: The changes are marked as 8, which are highlighted in yellow color on Page 10.

Point 5: As far as I'm concerned, the authors used the trained model to optimize the airfoil shape to achieve the best lift-drag characteristics. First of all, what kind of optimization algorithms was used? Moreover, I'm a little confused by the optimization setup. What is the purpose of this type of optimization? In my opinion, it fits best with morphing airfoils. For fixed shape ones, should we care more about the overall performance in the entire range, for example, AOA from 0 to 10 degrees, instead of the performance at a specific AOA?

Reply: Thank you for your suggestion. The rotor structure has a twist angle of -10° and the change of angle is linear along the spanwise direction, as shown in Fig.11. Along the spanwise direction, the rotor is divided into 20 sections, and the twist angle of the each cross section is corresponds to the parameter α. The shape for each section is optimized according to the parameter m, p, t which determine the section shape, as shown in Fig.2. The geometrical parameters of the two-dimensional airfoil for each cross-section are derived from the optimization results in Table.4, which means that we select the optimized airfoil parameters m, p, t which determine the section shape according to the angles α of different cross-sections. The SVM-G regression algorithms are used for Cl/Cd and the ANN 40-40 is used for Cl1.5/Cd according to the MSE and R2 score. Finally, the three-dimensional structure of Mars helicopter rotor with airfoil shape variation is obtained by the envelope method the best lift-drag characteristics, as shown in Fig.11. The AOA of the rotor varies linearly at different sections and the final envelope of airfoil is obtained by optimizing the section parameters at the corresponding AOA.

Change: The changes are marked as 9, which are highlighted in yellow color on Page 11.

Point 6: Similar to the question just mentioned above, I assume that the authors did the experiments for the optimized airfoil shape. As the optimal shapes for different AOA are different, which shape is used in the experiments? Or the airfoil shape changes during the experiments.

Reply: Thank you for your suggestion. More details are needed about the optimized airfoil shape, this work does not optimize for multiple airfoil types, but for specific airfoil type. The rotor structure has a twist angle of -10° and the change of angle is linear along the spanwise direction, as shown in Fig.11. Along the spanwise direction, the rotor is divided into 20 sections, and the twist angle of the each cross section is corresponds to the parameter α. The shape for each section is optimized according to the parameter m, p, t which determine the section shape, as shown in Fig.2. The AOA of the rotor varies linearly at different sections and the final envelope of airfoil is obtained by optimizing the section parameters at the corresponding AOA.

Change: The changes are marked as 9, which are highlighted in yellow color on Page 11.

Point 7: In the conclusion, the authors stated that ``The support vector machine-linear (SVM-L) regression algorithm exhibits relatively high prediction accuracy.''  As far as I'm concerned, in neither the Cl-Cd nor the Cl^1.5/Cd model, SVM-L is not considered superior, I'm wondering how the author drew the conclusion.

Reply: Thank you for your careful review. There is a writing error in the conclusion section, and the SVM-G algorithm exhibits smaller residuals for the prediction of airfoil lift resistance characteristics, achieving more accurate regression and prediction. The corresponding part in the conclusion of the revised manuscript has been corrected.

Change: The changes are marked as 10, which are highlighted in yellow color on Page 16.

Reviewer 2 Report

The paper presents a combination of ML-based, computation, and experimental studies for the optimization of the airfoil parameters, operating in Martian conditions. While the questions posed are interesting and the authors present a detailed description of the problem, I still have some concerns before recommending the publication of this manuscript. They are listed below.

1. The optimization aspect of the study is not clear to me. The authors come up with an ML model that predicts the lift-to-drag ratio. How do they use this model in section 3.4? What parameters are being optimized? How do they use these parameters in section 4 onward?

2. Another key information missing in the results section showing outputs of the ML models is which model is the best. The authors do not comment on this in the conclusion section as well. 

3. Are the optimized airfoil designs used in the numerical simulation and experimental study of the rotor in section 4? If yes, they should explicitly mention how and which airfoil design. If not, they should mention that explicitly as well. 

4. For section 2.1, why do authors use the 2D numerical simulation for airfoils? In the real-world operating scenario, I believe that the effect of three-dimensionality will kick in. If they are making an assumption of two-dimensional flow, they should clearly mention that and also the limitations of this assumption.

5. SST k-omega is a turbulence model. In two-dimensional simulations, I don't think there will be any turbulence. Is this RANS model even needed in that case?

6. Validation of their numerical simulations or the grid convergence study is missing from the paper. They should include at least one of these two, preferably both. 

7. Where is 'y' in equation 3? 

The quality of English in the paper is fair. There are minor issues but I believe they will be fixed by the editorial office in the proof-reading stage. 

Author Response

Reviewer #2: The paper presents a combination of ML-based, computation, and experimental studies for the optimization of the airfoil parameters, operating in Martian conditions. While the questions posed are interesting and the authors present a detailed description of the problem, I still have some concerns before recommending the publication of this manuscript. They are listed below.

Point 1: The optimization aspect of the study is not clear to me. The authors come up with an ML model that predicts the lift-to-drag ratio. How do they use this model in section 3.4? What parameters are being optimized? How do they use these parameters in section 4 onward?

Reply: Thank you for your careful review. More details are needed about the optimized airfoil shape, the rotor structure has a twist angle of -10° and the change of angle is linear along the spanwise direction, as shown in Fig.11. Along the spanwise direction, the rotor is divided into 20 sections, and the twist angle of the each cross section is corresponds to the parameter α. The shape for each section is optimized according to the parameter m, p, t which determine the section shape, as shown in Fig.2. The geometrical parameters of the two-dimensional airfoil for each cross-section are derived from the optimization results in Table.4, which means that we select the optimized airfoil parameters m, p, t which determine the section shape according to the angles α of different cross-sections. The SVM-G regression algorithms are used for Cl/Cd and the ANN 40-40 is used for Cl1.5/Cd according to the MSE and R2 score. Finally, the three-dimensional structure of Mars helicopter rotor with airfoil shape variation is obtained by the envelope method the best lift-drag characteristics, as shown in Fig.11.

Change: The changes are marked as 9, which are highlighted in yellow color on Page 11.

Point 2: Another key information missing in the results section showing outputs of the ML models is which model is the best. The authors do not comment on this in the conclusion section as well.

Reply: Thank you for your careful review. The SVM-G regression algorithms are used for Cl/Cd and the ANN 40-40 is used for Cl1.5/Cd according to the MSE and R2 score. The corresponding part are stated explicitly in section 3.4 and conclusion of the revised manuscript.

Change: The changes are marked as 8,10, which are highlighted in yellow color on Page 10,16.

Point 3: Are the optimized airfoil designs used in the numerical simulation and experimental study of the rotor in section 4? If yes, they should explicitly mention how and which airfoil design. If not, they should mention that explicitly as well.

Reply: Thank you for your careful review. The optimized airfoil designs used in the numerical simulation and experimental study of the rotor in section 4. The geometrical parameters of the two-dimensional airfoil for each cross-section are derived from the optimization results in Table.4, which means that we select the optimized airfoil parameters m, p, t which determine the section shape according to the angles α of different cross-sections. The final optimized airfoil model is fitted by applying the parameters (at different twist angle) cross-sections of the same airfoil for the experiment in section 4.

Change: The changes are marked as 9, which are highlighted in yellow color on Page 11.

Point 4: For section 2.1, why do authors use the 2D numerical simulation for airfoils? In the real-world operating scenario, I believe that the effect of three-dimensionality will kick in. If they are making an assumption of two-dimensional flow, they should clearly mention that and also the limitations of this assumption.

Reply: Thank you for your careful review. Note that the 3D structure of the rotor can also be obtained by 3D simulation, but considering the time cost of mesh drawing and simulation calculations, this work focuses on using a combination of 2D airfoil simulation and machine learning to optimize the design of the airfoil structure. Since the Mars atmosphere is complex, and the input parameters of 3D simulation are more than those of 2D simulation, 3D CFD simulation will greatly increase the computation time and convergence difficulty, and it still has influence on machine learning calculation. In future work we will further explore the 3D simulation. The reasons for choosing the two-bit emulation form are explained in detail and the corresponding part in the revised manuscript has been modified.

Change: The changes are marked as 4, which are highlighted in yellow color on Page 7.   

Point 5: SST k-omega is a turbulence model. In two-dimensional simulations, I don't think there will be any turbulence. Is this RANS model even needed in that case?

Reply: Thank you for your careful review. The Reynolds number of the flow field around the Martian ranges from 104 to 105, which causes the laminar flow in the airfoil boundary layer to be easily disturbed and transformed into turbulent flow. The laminar flow separation and laminar-turbulent transition mechanism of airfoil flow in the Martian atmosphere has been discussed in detail in work “Improved Mars helicopter aerodynamic rotor model for comprehensive analyses” and “Rotor propulsion modeling for low Reynolds number flow (Re < 105) for Martian rotorcraft flight”. Thus, the SST k-ω model is employed to simulate the turbulent flow on the airfoil surface in this paper, which is also used in the literature “Numerical study of unconventional airfoils at low Reynolds number for the application of Mars flight” to conduct the Martian airfoil aerodynamic performance simulation.

[1] doi.org/10.2514/1.J058045

[2] doi.org/10.2514/6.2022-3958

[3] dx.doi.org/10.13140/RG.2.2.33666.20163

Change: The changes are marked as 1, which are highlighted in yellow color on Page 4.

Point 6: Validation of their numerical simulations or the grid convergence study is missing from the paper. They should include at least one of these two, preferably both.

Reply: The reviewer's suggestions are very professional. In numerical simulations and CFD simulations, the discussion on the convergence part is not detailed enough. In this work, the finite element simulation is calculated until the residual is less than 10^-5, at which point the lift resistance coefficient converges to a stable value, at which point the calculation is considered converged.

Change: The changes are marked as 3, which are highlighted in yellow color on Page 7.

Point 7: Where is 'y' in equation 3?

Reply: Thank you for your careful review. There is a writing error in the formula. The corresponding part in the revised manuscript has been corrected.

Change: The changes are marked as 2, which are highlighted in yellow color on Page 6.

Round 2

Reviewer 1 Report

The authors have addressed most of my concerns, I suggest its publication in its present form. 

Reviewer 2 Report

The authors have addressed my concerns. I recommend the publication of the manuscript.